# Method for Remote Determination of Object Coordinates in Space Based on Exact Analytical Solution of Hyperbolic Equations

**DOI:** 10.3390/s20195472

**Published:** 2020-09-24

**Authors:** Vladimir Kuptsov, Vladimir Badenko, Sergei Ivanov, Alexander Fedotov

**Affiliations:** Institute of Physics, Nanotechnology and Telecommunications, Peter the Great Saint-Petersburg Polytechnic University, 29 Polytechnicheskaya str., 195251 Saint Petersburg, Russia; vdkuptsov@yandex.ru (V.K.); ivanov_si@spbstu.ru (S.I.); afedotov@spbstu.ru (A.F.)

**Keywords:** positioning, unmanned aerial vehicle, time difference of arrival, ambiguity, determination of 3D coordinates, desynchronization in time, TDoA fluctuations, local positioning system (LPS), global positioning system (GPS)

## Abstract

Accurate remote determination of the object coordinates in 3D space is one of the main questions in many applications. In one of the most popular methods, such determination of the location of an object uses the measurement by receiving an electromagnetic signal transmitted by several spatially distributed base stations (BS). The main problem is that it is necessary to reduce errors and computation time. To overcome these difficulties, an analytical method for determining the position of an object based on the analysis of time difference of arrival (TDoA) of signals from the transmitter of the object to the receivers of the BS is proposed. One of the main advantages of this method is that it is possible to eliminate the ambiguity in determining the coordinates of the object in space and to increase the accuracy of determining the coordinates when the TDoA measurement between base stations fluctuates. Applications for autonomous automotive vehicles and space-based positioning systems are analyzed. The results obtained show that the proposed algorithm has an accuracy of determining coordinates several times higher than the method of linearization of hyperbolic equations and is less sensitive to TDoA fluctuations at base stations.

## 1. Introduction

Many modern applications need accurate 2D/3D object coordinates, which are determined by remote sensing methods. These applications include, for example, driving unmanned vehicles or security systems in “smart city” projects [1,2]. In one of the most popular methods, such determination of the location of an object uses measurements by receiving an electromagnetic signal transmitted by several spatially distributed base stations, taking into account shapes and other properties of objects [3]. Commonly used measurements are time delay for active scenarios [4] and time differences for passive environments [5]. The main aim of positioning tasks is to determine the coordinates (position) of the object/target, which can be carried out using various algorithms [6,7] and the most popular of them are discussed as follows.

For example, in [8], a control algorithm for an unmanned aerial vehicle (UAV) to circumnavigate an unknown target at a fixed radius when the UAV is unable to determine its location based on measurement errors and modeling the system by stochastic differential equations. Among other algorithms, it can be pointed out that with RSSI (Received Strength Signal Indication), the distance to the object is estimated by the signal power, but this algorithm usually has successful application for indoor positioning with a wireless network [9,10,11]. The algorithm AoA (Angle of Arrival) is based on the location of the object by determination within the triangle area formed by the intersection of the axes of the antenna patterns of the sectors of the three base stations (modified triangulation method) and usually plays a role as a part of another target tracking algorithm, for example, 3D pseudolinear Kalman filter (KF) algorithm [12,13,14] or 3D source localization method for acoustic sensor networks [15]. The algorithm ToF (Time of Flight) is based on measuring the travel time of an electromagnetic wave from an object transmitter to a base station by using a signal with linear frequency modulation [16,17,18]. This algorithm is very popular for applications in medicine [19] but also has application in unmanned aerial vehicle navigation [20]. Close in meaning to ToF is the algorithm ToA (Time of Arrival), which is based on distance calculation by measuring the signal transit time from the mobile terminal to the base station as the difference between the time the signal was sent and received [21,22,23,24]. The algorithm ToA is often used with TDoA (Time Difference of Arrival) [25] and FDoA (Frequency difference of arrival) [26], which are also often used together [27,28,29,30]. TDoA is a very popular algorithm for different applications and measures the difference in the time of arrival of a signal from an object to several base stations [31,32,33]. FDoA, which is also named DD (Differential Doppler), is a method analogous to TDoA for estimating the location of a radio emitter based on observations from other points according to different Doppler shift observations of the emitter at different locations [34,35]. Very similar to ToA is RTT (Round Trip Time) method, in which a base station also sends a signal to a mobile device and waits for a response signal [36,37,38].

The TDoA method is highly accurate, and one of its main features is that it requires time synchronization only at base stations [39,40,41,42,43]. If one analyzes the surfaces on which the difference in the arrival times of the microwave signal in TDoA method to two base stations (BS) is constant, one finds that they are hyperboloids in space with both BSs in their focus [39,44]. In such cases, the coordinates of the target can be identified as an intersection of these surfaces using, for example, the Taylor-series method at a reasonable noise level [45,46,47]. But this Taylor-series method is an iterative method, and it strongly depends on an initial assumption (starting point) and requires large computational resources [48]. For solving such problems in [47], it was proposed to use Taylor series expansion for linearization of hyperbolic equations, which leads to a more simple matrix form of equations [49,50].

A linear frequency-modulated microwave radar waveform is used in many applications, including microwave-range automobile radars for the simultaneous measurement of velocity and distance, which use signals of a special shape, including frequency-modulated continuous-wave (FMCW) [51]. Mathematical models of the received microwave signal for automotive radar systems taking into account a different range of velocity and distance for vehicles, were proposed in [52,53,54,55,56,57,58]. In [52], it is shown that the use of phase methods to determine the speed of the target significantly reduces the noise immunity of the radar, and methods based on processing the amplitude spectrum of the received radio signal by the Fast Fourier Transformation (FFT) are therefore preferable. In [53] is the algorithm for estimating the speed and distance to the target as applied to the automotive radar with linear FMCW, using the FFT for amplitude spectrum and taking into account the occurrence of false targets. The results of synthesis and analysis of the effectiveness of the optimal location-based system for joint detection and estimation of informative parameters of quasi-determined radar signals with frequency modulated continuous wave are presented in [56]. In [57], the authors explore the properties of the multi-target method for small unmanned vehicle parameters’ remote determination by microwave radars, which allows significantly expanding the range of unambiguously determined speeds of unmanned vehicles. The reasons for the occurrence of false targets and their number in the mode of the parameters simultaneous measurement for the set of unmanned vehicles are identified. Using a computer experiment in LabVIEW (Laboratory Virtual Instrumentation Engineering Workbench), the probability of the unmanned vehicles’ true parameters determination at various ratios of the signal to noise at the input of the radar receiver in multi-target mode was investigated. The influence of millimeter-wave radar receiver noise on the probability of unambiguous determination of unmanned vehicles’ speed and range in the intelligent transportation system of the “smart city” is investigated in [58]. For the proposed new multi-target detection method for FMCW radar, the effect of the technical parameters of the vehicle radars on the required signal-to-noise ratio (SNR) of the receiver is estimated to ensure the probability of true determination of target parameters at 98%.

An exact analytical solution of equations for the task of localization of an object using a number of sensors is often challenged by outlier observations when the number of TDoA measurements is equal to the number of unknown transmitter coordinates [59,60,61]. The measurement equations are nonlinear, and an exact algebraic solution exists only in the specific case of a predetermined scenario in which the number of measurements is equal to the number of unknowns. The solution in [60] cannot use TDoA measurements with additional base stations improving positioning accuracy, and it has a high computational requirement for solving nonlinear equations. Compared to the solutions in [60], the solution proposed in [61] is more computationally attractive, requires only the roots of a quadratic equation, is more general and robust, and can work with arbitrary sensor arrangements.

A method proposed in [62] is based on Spherical-Intersection using only the root of a quadratic equation, but this method is not robust and fails to produce a reasonable solution for some sensor arrangements, due to the need to invert a matrix with the coordinates of sensors. Exact solutions can also be found in [63,64,65], and authors sometimes formulated an appropriated optimization task in form of a polynomial system, the solution of which was found by numerical algebraic methods or polynomial continuation techniques.

A good description of analytical methods is presented in [66,67,68]. Transformation of the coordinates of a hyperbola during a shift and rotation on a plane can be found in [66]. According to this transformation in [67], the effectiveness of two methods of TDOA analytical solution was compared based on the task of finding intersection points of hyperboloids (possible positions of a target). The first method analyzed was based on coordinate transformation from the initial system to a new system to simplify equations solving, and the second one was based on matrix. In [68], the results of an experimental verification of an analytical method based on coordinate transformation are presented. Finally, in [69], a high-precision analytical TDoA algorithm for determining the coordinates of an object on a plane (2D model) is proposed for eliminating the ambiguity of coordinates determination. Of great interest are recent works especially for indoor and AoA localization [70,71,72,73].

In this paper, an analytical method for determining the position of a target based on the analysis of TDoA of microwave radar signals from the transmitter to Base Stations (BS) receivers is proposed. The main feature of the method is that it can eliminate the ambiguity in determining of 3D coordinates of a target and improves the accuracy of determining coordinates when the TDoA measurements fluctuate on BS.

The materials of the article are presented in the following form: In Section 2.1, the fundamentals of the linearization method are presented without going into the specifics of solving linear equations by matrix methods. Section 2.2 sets out the proposed analytical method for determining the coordinates of an object in a positioning system. Section 3.1 describes the spatial ambiguity problem inherent in simple analytical methods. A comparison of the accuracy of determining coordinates by the linearization method and the proposed method at various levels of TDoA fluctuations for local and global positioning systems are carried out in Section 3.2. The results are discussed in Section 4. The paper ends with conclusions (Section 5) from the work done.

## 2. Materials and Methods

### 2.1. Linearization Method of Hyperbolic Equations

The method of linearizing hyperbolic equations was first proposed in [47]. The widespread use of this method at the present time is explained by the fact that it allows one to avoid solving nonlinear equations, which means that significant computing power of the positioning system processor is not required. The method consists in transforming nonlinear equations into a set of linear equations Ax = b in matrix form and further solving the system of equations by matrix methods [25,46,47,50,61,65]. We will compare our proposed algorithm (Section 2.2) with the linearization method; therefore, we will briefly outline the basics of the linearization method without delving into the specifics of solving linear equations by matrix methods.

In the absence of errors in measuring TDoA and receiving line-of-sight (LOS) signals, the real values of the difference in the arrival times of the TDoA signal between BS*_i_* and BS_1_ are determined by the expression:(1)$$\Delta \tau_{i1}=\frac{r_{i}-r_{1}}{c_{l}}=\frac{r_{i1}}{c_{l}},\;i=2,\ldots ,M$$
where *M*—amount of BS, *c_l_*—speed of light, $r_{i}=\sqrt{{\left(x_{i}-x\right)}^{2}+{\left(y_{i}-y\right)}^{2}+{\left(z_{i}-z\right)}^{2}}$—distance between object and BS*_i_*, *r*_1_—distance between object and BS_1_, [*x_i_, y_i_, z_i_*]—BS*_i_* coordinates, [*x, y, z*]—object coordinates. From $r_{i1}=r_{i}-r_{1}$ follows $r_{i1}+r_{1}=r_{i}$. Substitution of the coordinates of the Cartesian system in Equation (1), squaring the right and left sides of the equation, expansion in a Taylor series, and linearization leads to the equation:(2)$$\begin{array}{c} \left(x_{i}-x_{1}\right)\left(x-x_{1}\right)+\left(y_{i}-y_{1}\right)\left(y-y_{1}\right)+\left(z_{i}-z_{1}\right)\left(z-z_{1}\right)+c_{l}\cdot \Delta \tau_{i1}\cdot r_{1}=\ldots \\ \ldots =\frac{1}{2}\left({\left(x_{i}-x_{1}\right)}^{2}+{\left(y_{i}-y_{1}\right)}^{2}+{\left(z_{i}-z_{1}\right)}^{2}-{\left(c_{l}\cdot \Delta \tau_{i1}\right)}^{2}\right), \end{array}$$
which in matrix form has the form ***Aθ** = **b***, where
(3)$$\begin{array}{c} A=\left[\begin{array}{cccc} x_{2}-x_{1} & y_{2}-y_{1} & z_{2}-z_{1} & c_{l}\cdot \Delta \tau_{21}\\ x_{3}-x_{1} & y_{3}-y_{1} & z_{3}-z_{1} & c_{l}\cdot \Delta \tau_{31}\\ \cdot & \cdot & \cdot & \cdot \\ x_{M}-x_{1} & y_{M}-y_{1} & z_{M}-z_{1} & c_{l}\cdot \Delta \tau_{M1} \end{array}\right],\;\theta ={\left[\begin{array}{cccc} x-x_{1} & y-y_{1} & z-z_{1} & r_{1} \end{array}\right]}^{T}\\ b=\frac{1}{2}\left[\begin{array}{c} {\left(x_{2}-x_{1}\right)}^{2}+{\left(y_{2}-y_{1}\right)}^{2}+{\left(z_{2}-z_{1}\right)}^{2}-{\left(c_{l}\cdot \Delta \tau_{21}\right)}^{2}\\ {\left(x_{3}-x_{1}\right)}^{2}+{\left(y_{3}-y_{1}\right)}^{2}+{\left(z_{3}-z_{1}\right)}^{2}-{\left(c_{l}\cdot \Delta \tau_{31}\right)}^{2}\\ \cdot \\ {\left(x_{M}-x_{1}\right)}^{2}+{\left(y_{M}-y_{1}\right)}^{2}+{\left(z_{M}-z_{1}\right)}^{2}-{\left(c_{l}\cdot \Delta \tau_{M1}\right)}^{2} \end{array}\right] \end{array}$$

The superscript [ ]*^T^* denotes a matrix transposition operation. The solution of the equation ***Aθ*** = ***b*** can be carried out in various ways of solving the system of linear equations. After calculating the values of the vector ***θ***, the coordinates of the target are determined $x=\theta_{1}+x_{1}$; $y=\theta_{2}+y_{1}$; $z=\theta_{3}+z_{1}$; where *x*_1_, *y*_1_, *z*_1_—known coordinates of BS_1_.

### 2.2. The Proposed Analytical Method for Determining the Coordinates of an Object in a Positioning System with 5 Base Stations

The variant of building a positioning system in space with 4 BSs is the most economically profitable; however, it has a disadvantage that the intersection of two hyperboloids can occur along two lines in space. In this case, it becomes impossible to identify the true position of the object. To eliminate the ambiguity in determining the position of the object, it is necessary to include the fifth BS in the positioning system. With the addition of the fifth BS, it becomes possible to solve four systems of equations, which will have one common root corresponding to the true position of the target. An analytical solution to the problem of determining the location of an object in 3D for a positioning system of 4 BSs was obtained in [67], but in [67], not all values of the coefficients are given, which complicates the direct use of the algorithm. In addition, in [67], the problem of eliminating ambiguity in determining the coordinates was not solved.

In what follows, the development of an analytical algorithm that eliminates the spatial zones of ambiguity in determining the coordinates of the target in space with high accuracy coordinates determination of the object is presented.

Let us consider a spatial positioning system that includes five microwave signal receivers BC*_C_*, BS*_L_*, BS*_R_*, BS*_U_*, BS*_D_* (Figure 1).

The coordinates of the transmitter position are solutions to the system of hyperbolic equations:(4)$$\left\{ \begin{array}{l} \sqrt{{\left(x-x_{L}\right)}^{2}+{\left(y-y_{L}\right)}^{2}+{\left(z-z_{L}\right)}^{2}}-\sqrt{{\left(x-x_{C}\right)}^{2}+{\left(y-y_{C}\right)}^{2}+{\left(z-z_{C}\right)}^{2}}=c\cdot \Delta \tau_{LC}\\ \sqrt{{\left(x-x_{R}\right)}^{2}+{\left(y-y_{R}\right)}^{2}+{\left(z-z_{R}\right)}^{2}}-\sqrt{{\left(x-x_{C}\right)}^{2}+{\left(y-y_{C}\right)}^{2}+{\left(z-z_{C}\right)}^{2}}=c\cdot \Delta \tau_{RC}\\ \sqrt{{\left(x-x_{U}\right)}^{2}+{\left(y-y_{U}\right)}^{2}+{\left(z-z_{U}\right)}^{2}}-\sqrt{{\left(x-x_{C}\right)}^{2}+{\left(y-y_{C}\right)}^{2}+{\left(z-z_{C}\right)}^{2}}=c\cdot \Delta \tau_{UC}\\ \sqrt{{\left(x-x_{D}\right)}^{2}+{\left(y-y_{D}\right)}^{2}+{\left(z-z_{D}\right)}^{2}}-\sqrt{{\left(x-x_{C}\right)}^{2}+{\left(y-y_{C}\right)}^{2}+{\left(z-z_{C}\right)}^{2}}=c\cdot \Delta \tau_{DC} \end{array}\right.$$
where ∆*τ_LC_*, ∆*τ_RC_*, ∆*τ_UC_*, ∆*τ_DC_*—TDoA of the microwave signal between BS*_L,R,U,D_* and reference station BS*_C_*.

After substitution of *X = x* − *x_C_*, *Y = y* − *y_C_*, *Z = z* − *z_C_* the system of equations can be rewritten as:(5)$$\begin{array}{c} \left\{ \begin{array}{l} \sqrt{{\left(X-X_{L}\right)}^{2}+{\left(Y-Y_{L}\right)}^{2}+{\left(Z-Z_{L}\right)}^{2}}-\sqrt{X^{2}+Y^{2}+Z^{2}}=L\\ \sqrt{{\left(X-X_{R}\right)}^{2}+{\left(Y-Y_{R}\right)}^{2}+{\left(Z-Z_{R}\right)}^{2}}-\sqrt{X^{2}+Y^{2}+Z^{2}}=R\\ \sqrt{{\left(X-X_{U}\right)}^{2}+{\left(Y-Y_{U}\right)}^{2}+{\left(Z-Z_{U}\right)}^{2}}-\sqrt{X^{2}+Y^{2}+Z^{2}}=U\\ \sqrt{{\left(X-X_{D}\right)}^{2}+{\left(Y-Y_{D}\right)}^{2}+{\left(Z-Z_{D}\right)}^{2}}-\sqrt{X^{2}+Y^{2}+Z^{2}}=D \end{array}\right.,\;\mathrm{where}\;\\ \begin{array}{l} X_{L}=x_{L}-x_{C},\;Y_{L}=y_{L}-y_{C}{,\;Z}_{L}=z_{L}-z_{C},\;L=c\cdot \Delta \tau_{LC}\\ X_{R}=x_{R}-x_{C},\;Y_{R}=y_{R}-y_{C}{,\;Z}_{R}=z_{R}-z_{C},\;R=c\cdot \Delta \tau_{RC}\\ X_{U}=x_{U}-x_{C},\;Y_{U}=y_{U}-y_{C}{,\;Z}_{U}=z_{U}-z_{C},\;U=c\cdot \Delta \tau_{UC}\\ X_{D}=x_{D}-x_{C},\;Y_{D}=y_{D}-y_{C}{,\;Z}_{D}=z_{D}-z_{C},\;D=c\cdot \Delta \tau_{DC} \end{array} \end{array}$$

Let us denote $K=\sqrt{X^{2}+Y^{2}+Z^{2}}$, where *K >* 0:(6)$$K^{2}=X^{2}+Y^{2}+Z^{2},$$

The system of Equations (5) can be rewritten as:(7)$$\left\{ \begin{array}{l} \sqrt{{\left(X-X_{L}\right)}^{2}+{\left(Y-Y_{L}\right)}^{2}+{\left(Z-Z_{L}\right)}^{2}}=K+L\\ \sqrt{{\left(X-X_{R}\right)}^{2}+{\left(Y-Y_{R}\right)}^{2}+{\left(Z-Z_{R}\right)}^{2}}=K+R\\ \sqrt{{\left(X-X_{U}\right)}^{2}+{\left(Y-Y_{U}\right)}^{2}+{\left(Z-Z_{U}\right)}^{2}}=K+U\\ \sqrt{{\left(X-X_{D}\right)}^{2}+{\left(Y-Y_{D}\right)}^{2}+{\left(Z-Z_{D}\right)}^{2}}=K+D \end{array}\right.,$$

Squaring and reducing the general terms leads to the form:(8)$$\begin{array}{c} \left\{ \begin{array}{l} -2X_{L}X-2Y_{L}Y-2Z_{L}Z=2LK+L^{2}-X_{L}^{2}-Y_{L}^{2}-Z_{L}^{2}\\ -2X_{R}X-2Y_{R}Y-2Z_{R}Z=2RK+R^{2}-X_{R}^{2}-Y_{R}^{2}-Z_{R}^{2}\\ -2X_{U}X-2Y_{U}Y-2Z_{U}Z=2UK+U^{2}-X_{U}^{2}-Y_{U}^{2}-Z_{U}^{2}\\ -2X_{D}X-2Y_{D}Y-2Z_{D}Z=2DK+D^{2}-X_{D}^{2}-Y_{D}^{2}-Z_{D}^{2} \end{array}\right.,\;\mathrm{or}\;\\ \left\{ \begin{array}{l} -2X_{L}X-2Y_{L}Y-2Z_{L}Z=E+2LK\\ -2X_{R}X-2Y_{R}Y-2Z_{R}Z=F+2RK\\ -2X_{U}X-2Y_{U}Y-2Z_{U}Z=G+2UK\\ -2X_{D}X-2Y_{D}Y-2Z_{D}Z=H+2DK \end{array}\right., \end{array}$$
where $E=L^{2}-X_{L}^{2}-Y_{L}^{2}-Z_{L}^{2}$; $F=R^{2}-X_{R}^{2}-Y_{R}^{2}-Z_{R}^{2}$; $G=U^{2}-X_{U}^{2}-Y_{U}^{2}-Z_{U}^{2}$; $H=D^{2}-X_{D}^{2}-Y_{D}^{2}-Z_{D}^{2}$.

In matrix form, the four systems of equations are:(9)$$\begin{array}{l} \left[\begin{array}{ccc} -2X_{L} & -2Y_{L} & -2Z_{L}\\ -2X_{R} & -2Y_{R} & -2Z_{R}\\ -2X_{U} & -2Y_{U} & -2Z_{U} \end{array}\right]\cdot \left[\begin{array}{c} X\\ Y\\ Z \end{array}\right]=\left[\begin{array}{c} E+2LK\\ F+2RK\\ G+2UK \end{array}\right],\;\\ \left[\begin{array}{ccc} -2X_{R} & -2Y_{R} & -2Z_{R}\\ -2X_{U} & -2Y_{U} & -2Z_{U}\\ -2X_{D} & -2Y_{D} & -2Z_{D} \end{array}\right]\cdot \left[\begin{array}{c} X\\ Y\\ Z \end{array}\right]=\left[\begin{array}{c} F+2RK\\ G+2UK\\ H+2DK \end{array}\right],\;\\ \left[\begin{array}{ccc} -2X_{U} & -2Y_{U} & -2Z_{U}\\ -2X_{D} & -2Y_{D} & -2Z_{D}\\ -2X_{L} & -2Y_{L} & -2Z_{L} \end{array}\right]\cdot \left[\begin{array}{c} X\\ Y\\ Z \end{array}\right]=\left[\begin{array}{c} G+2UK\\ H+2DK\\ E+2LK \end{array}\right],\;\\ \left[\begin{array}{ccc} -2X_{D} & -2Y_{D} & -2Z_{D}\\ -2X_{L} & -2Y_{L} & -2Z_{L}\\ -2X_{R} & -2Y_{R} & -2Z_{R} \end{array}\right]\cdot \left[\begin{array}{c} X\\ Y\\ Z \end{array}\right]=\left[\begin{array}{c} H+2DK\\ E+2LK\\ F+2RK \end{array}\right] \end{array}$$

It should be noted that when using 4 BS*s* in the positioning system, there will be only one system of equations.

Solution of the first system of equations (*i* = 1) is:(10)$$\begin{array}{c} X_{i}=\frac{1}{\Delta_{i}}\cdot \left|\begin{array}{ccc} E+2LK & -2Y_{L} & -2Z_{L}\\ F+2RK & -2Y_{R} & -2Z_{R}\\ G+2UK & -2Y_{U} & -2Z_{U} \end{array}\right|=M_{iX}\cdot K+N_{iX},\;\mathrm{where}\;\\ M_{iX}=\frac{2}{\Delta_{i}}\cdot \left(L\Delta_{iX1}-R\Delta_{iX2}+U\Delta_{iX3}\right),\;N_{iX}=\frac{1}{\Delta_{i}}\cdot \left(E\Delta_{iX1}-F\Delta_{iX2}+G\Delta_{iX3}\right);\;\\ \Delta_{iX1}=\left|\begin{array}{cc} -2Y_{R} & -2Z_{R}\\ -2Y_{U} & -2Z_{U} \end{array}\right|=4\left(Y_{R}Z_{U}-Y_{U}Z_{R}\right),\;\\ \Delta_{iX2}=\left|\begin{array}{cc} -2Y_{L} & -2Z_{L}\\ -2Y_{U} & -2Z_{U} \end{array}\right|=4\left(Y_{L}Z_{U}-Y_{U}Z_{L}\right),\;\\ \Delta_{iX3}=\left|\begin{array}{cc} -2Y_{L} & -2Z_{L}\\ -2Y_{R} & -2Z_{R} \end{array}\right|=4\left(Y_{L}Z_{R}-Y_{R}Z_{L}\right) \end{array}$$
(11)$$\begin{array}{c} Y_{i}=\frac{1}{\Delta_{i}}\cdot \left|\begin{array}{ccc} -2X_{L} & E+2LK & -2Z_{L}\\ -2X_{R} & F+2RK & -2Z_{R}\\ -2X_{U} & G+2UK & -2Z_{U} \end{array}\right|=M_{iY}\cdot K+N_{iY},\;\mathrm{where}\;\\ M_{iY}=\frac{2}{\Delta_{i}}\cdot \left(-L\Delta_{iY1}+R\Delta_{iY2}-U\Delta_{iY3}\right),\;N_{iY}=\frac{1}{\Delta_{i}}\cdot \left(-E\Delta_{iY1}+F\Delta_{iY2}-G\Delta_{iY3}\right);\;\\ \Delta_{iY1}=\left|\begin{array}{cc} -2X_{R} & -2Z_{R}\\ -2X_{U} & -2Z_{U} \end{array}\right|=4\left(X_{R}Z_{U}-X_{U}Z_{R}\right),\;\\ \Delta_{iY2}=\left|\begin{array}{cc} -2X_{L} & -2Z_{L}\\ -2X_{U} & -2Z_{U} \end{array}\right|=4\left(X_{L}Z_{U}-X_{U}Z_{L}\right),\;\\ \Delta_{iY3}=\left|\begin{array}{cc} -2X_{L} & -2Z_{L}\\ -2X_{R} & -2Z_{R} \end{array}\right|=4\left(X_{L}Z_{R}-X_{R}Z_{L}\right) \end{array}$$
(12)$$\begin{array}{c} Z_{i}=\frac{1}{\Delta_{i}}\cdot \left|\begin{array}{ccc} -2X_{L} & -2Y_{L} & E+2LK\\ -2X_{R} & -2Y_{R} & F+2RK\\ -2X_{U} & -2Y_{U} & G+2UK \end{array}\right|=M_{iZ}\cdot K+N_{iZ},\;\mathrm{where}\;\\ M_{iZ}=\frac{2}{\Delta_{i}}\cdot \left(L\Delta_{iZ1}-R\Delta_{iZ2}+U\Delta_{iZ3}\right),\;N_{iZ}=\frac{1}{\Delta_{i}}\cdot \left(E\Delta_{iZ1}-F\Delta_{iZ2}+G\Delta_{iZ3}\right);\;\\ \Delta_{iZ1}=\left|\begin{array}{cc} -2X_{R} & -2Y_{R}\\ -2X_{U} & -2Y_{U} \end{array}\right|=4\left(X_{R}Y_{U}-X_{U}Y_{R}\right),\;\\ \Delta_{iZ2}=\left|\begin{array}{cc} -2X_{L} & -2Y_{L}\\ -2X_{U} & -2Y_{U} \end{array}\right|=4\left(X_{L}Y_{U}-X_{U}Y_{L}\right),\;\\ \Delta_{iZ3}=\left|\begin{array}{cc} -2X_{L} & -2Y_{L}\\ -2X_{R} & -2Y_{R} \end{array}\right|=4\left(X_{L}Y_{R}-X_{R}Y_{L}\right) \end{array}$$

The general determinant of the first system of equations is the following:(13)$$\Delta_{i}=\left|\begin{array}{ccc} -2X_{L} & -2Y_{L} & -2Z_{L}\\ -2X_{R} & -2Y_{R} & -2Z_{R}\\ -2X_{U} & -2Y_{U} & -2Z_{U} \end{array}\right|=-8\cdot \left(X_{L}Y_{R}Z_{U}+X_{U}Y_{L}Z_{R}+X_{R}Y_{U}Z_{L}-X_{L}Y_{U}Z_{R}-X_{R}Y_{L}Z_{U}-X_{U}Y_{R}Z_{L}\right)$$

The solutions of the second (*i* = 2), third (*i* = 3), and fourth (*i* = 4) systems of equations are obtained from the solution of the first system by successive replacement of the notation (Table 1):

Substitution of Equations (10)–(12) into Equation (6) defines a quadratic equation with respect to the variable *K*: $a_{i}K^{2}+b_{i}K+c_{i}=0$, where *i* corresponds to the choice of a pair of hyperboloids, *i =* 1, 2, 3, 4.
(14)$$a_{i}=M_{iX}^{2}+M_{iY}^{2}+M_{iZ}^{2}-1,\;b_{i}=2\left(M_{iX}N_{iX}+M_{iY}N_{iY}+M_{iZ}N_{iZ}\right),\;c_{i}=N_{iX}^{2}+N_{iY}^{2}+N_{iZ}^{2}$$

The roots of the quadratic equation are the following ($b_{i}{}^{2}-4a_{i}c_{i}\geq 0$):(15)$$K_{i1}=\frac{-b_{i}+\sqrt{b_{i}{}^{2}-4a_{i}c_{i}}}{2a_{i}},\;K_{i2}=\frac{-b_{i}-\sqrt{b_{i}{}^{2}-4a_{i}c_{i}}}{2a_{i}}$$

In the case that one of the two roots *K_i_*_1_ and *K_i_*_2_ is negative for the same value of *i*, it can be immediately excluded from the solution, and then another root of the equation remains in the algorithm. However, a situation is possible when both roots *K_i_*_1_ and *K_i_*_2_ are positive. In this case, it becomes impossible to determine the coordinates of the object. It is for such a fairly common case that the fifth BS has to be used in the 3D positioning system based on the analytical method.

Substitution of roots *K_i_*_1_ and *K_i_*_2_ in Equations (10)–(12) defines eight possible sets of *x*; *y*; *z* coordinates of an object, defining eight points in space:(16)$$\begin{array}{c} {}[x_{i1}=X_{i1}+x_{C},\;y_{i1}=Y_{i1}+y_{C},\;z_{i1}=Z_{i1}+z_{C}],\\ {}[x_{i2}=X_{i2}+x_{C},\;y_{i2}=Y_{i2}+y_{C},\;z_{i2}=Z_{i2}+z_{C}],\;\mathrm{where}\\ X_{i1}=M_{iX}\cdot K_{i1}+N_{iX},\;Y_{i1}=M_{iY}\cdot K_{i1}+N_{iY},\;Z_{i1}=M_{iZ}\cdot K_{i1}+N_{iZ}\\ X_{i2}=M_{iX}\cdot K_{i2}+N_{iX},\;Y_{i2}=M_{iY}\cdot K_{i2}+N_{iY},\;Z_{i2}=M_{iZ}\cdot K_{i2}+N_{iZ} \end{array}$$

In the absence of TDoA measurement errors from eight calculated sets of *x*; *y*; *z* -coordinates, the coordinates of four points will be the same. It is this decision that is the true decision. To identify it, we carry out the following steps of the Algorithm 1:
**Algorithm 1.** Identifying of the true solutionCalculation of 48 = (8 points × 6 combinations) distances *D* between points-candidates for the true value, determined by Equation (16):$D_{i1\_j1}=\sqrt{{\left(x_{i1}-x_{j1}\right)}^{2}+{\left(y_{i1}-y_{j1}\right)}^{2}+{\left(z_{i1}-z_{j1}\right)}^{2}},$$D_{i2\_j2}=\sqrt{{\left(x_{i2}-x_{j2}\right)}^{2}+{\left(y_{i2}-y_{j2}\right)}^{2}+{\left(z_{i2}-z_{j2}\right)}^{2}}$where both indices *i* and *j* take the values 1, 2, 3, 4, and i≠j.Sorting the elements of a string consisting of six distances related to one candidate point in ascending order.Summing the first three elements of the row.For all eight candidate points, writing the sum of the first three elements to a line. The meaning of each sum is the sum of the distances between the three nearest points.In parallel with step 4 of the algorithm, forming three lines by *x*; *y*; *z* from eight elements in each line. Each of the eight line elements corresponds to the coordinates of the candidate points.Making a cycle on 5 configurations of the spatial arrangement of BS. That is, each BS once becomes a reference. For each configuration, its own line of eight “sums of minimum distances” is formed. The strings are combined into a matrix.In the resulting matrix of “sums of minimum distances”, seeking the element with the minimum value and its indices in the matrix.In parallel with item 7, forming three coordinate matrices from the rows of item 5. Each row of the matrix corresponds to the configuration of the location of the BS.Selecting, from the matrix of item 8 by the indices of item 7, the coordinates of the estimation of the position of the object $\bar{x}$; $\bar{y}$; $\bar{z}$. The selected coordinates are the closest to the true coordinates of the target. The rest of the coordinates are either false or determine the coordinates of the target with a greater error.

## 3. Results

### 3.1. Spatial Ambiguity Problem

In [67], an example of calculating target coordinates is given, in which the values of *K_i_*_1_ and *K_i_*_2_ are both positive. Two variants of target coordinates were obtained. The authors chose one of the options based on the disposition of the system of receivers to determine final target coordinates; however, they did not specify what kind of disposition it is. Most likely, it meant the negative *z* coordinate of the target, which means height. It should be noted that *K_i_*_1_ and *K_i_*_2_ are both positive and often occur with a positive *z* coordinate. In this case, the algorithm [56] is generally unable to give the correct values of the target coordinates. The method of linearizing hyperbolic equations also leads to a gross error in determining the coordinates of the object. In this article, a method is proposed that eliminates the ambiguity of determining the coordinates.

Figure 2 shows the areas in which *K_i_*_1_ and *K_i_*_2_ are both positive, and therefore, there is no way to determine the true value of the target coordinates. The calculations were carried out at BS coordinates (−40; 40; 0), (40; 40; 10), (40; −40; 20), (−40; −40; 30) m. In Figure 2, sensor projections on the horizontal plane are marked with blue dots. The target height took the values 0 m (a); 15 m (b); 30 m (s). If the heights of the sensors differ slightly, the undetectable areas expand significantly (*z_C,L,R,U_* = 8; 10; 12; 14 m, Figure 2d). With the same BS heights, the target positions are not determined anywhere using the algorithm [67]. Significant areas, where the target coordinates are not determined, exist for almost all values of the BS coordinates.

The method proposed by us is free from the indicated problem of ambiguity in determining the coordinates of an object. The method allows you to determine the coordinates of the object at all points in Figure 2.

### 3.2. Influence of the TDoA Fluctuations on the Accuracy of Coordinate Estimation

TDoA measurement errors occur mainly due to time desynchronization at base stations, as well as the conditions of propagation and reception of radio signals, and are the main reason for inaccurate determination of the object’s position. Using a computer experiment in the LabVIEW environment, we investigated the dependence of the root mean square (RMS) deviation of the object coordinates on the standard deviation of the TDoA of microwave signals in the positioning radar. For this, Gaussian white noise with the same standard deviation value for all BSs was added to the TDoA value at the input of each receiver of the positioning system. To determine the standard deviation of the object position estimate, n = 500 computational experiments were carried out for each value of the standard deviation of the Gaussian white noise. The standard deviation of the object coordinates estimate was determined in accordance with the expression:(17)$$RMS=\sqrt{\frac{1}{n}\sum_{k=1}^{n}\left[{\left({\bar{x}}_{k}-x\right)}^{2}+{\left({\bar{y}}_{k}-y\right)}^{2}+{\left({\bar{z}}_{k}-z\right)}^{2}\right]},$$
where index *k*—experiment number, [${\bar{x}}_{k}$;${\bar{y}}_{k}$;${\bar{z}}_{k}$] —estimation of object coordinates in *k* experiment, [*x*; *y*; *z*]—object’s true coordinates.

Figure 3 (time is in picoseconds) shows the dependences of the RMS deviation of the object coordinates estimate on the standard deviation of the Gaussian white noise of the difference in the arrival times of microwave signals to the BS for the matrix linearization algorithm and the proposed analytical algorithm. Dependents were calculated for object coordinates [*x*; *y*; *z*] = [10; 1; 5] and coordinates of BS*_C_* [*x_C_*; *y_C_*; *z_C_*] = [−40; 40; 0], BS*_L_* [*x_L_*; *y_L_*; *z_L_*] = [40; 40; 5], BS*_R_* [*x_R_*; *y_R_*; *z_R_*] = [40; −40; 10], BS*_U_* [*x_U_*; *y_U_*; *z_U_*] = [−40; −40; 15] and BS*_D_* [*x_D_*; *y_D_*; *z_D_*] = [40; 40; 20] m.

From the simulation results, it follows that the proposed algorithm provides the accuracy of determining the coordinates of the object 10 times higher than the method of linearization of hyperbolic equations. The proposed method works with high accuracy (5 cm) with BS standard deviation of TDoA up to 200 ps, while the linearization method with such BS standard deviation of TDoA provides an accuracy of 60 cm. Gross failures in the linearization algorithm start at 500 ps standard deviation of TDoA, while the proposed analytical algorithm works without gross failures at 1000 ps standard deviation of TDoA.

The following RMS dependences of the coordinate estimates were obtained for the configuration of five BSs located on a circle as in [74] with a radius of 50 m, which corresponds to the Local Positioning System (LPS). Coordinates were as follows: BS*_C_* [*x_C_*; *y_C_*; *z_C_*] = [29.4; −40.45; 20], BS*_L_* [*x_L_*; *y_L_*; *z_L_*] = [−29.4; −40.45; 25], BS*_R_* [*x_R_*; *y_R_*; *z_R_*] = [−47.55; 15.45; 30], BS*_U_* [*x_U_*; *y_U_*; *z_U_*] = [0; 50; 0], and BS*_D_* [*x_D_*; *y_D_*; *z_D_*] = [47.55; 15.45; 10] m, which is presented in Figure 4. Target height is z = 5 m. The number of computational experiments n = 100.

It should be noted that the algorithm with one equation [56], as well as the linearization method, produces a gross error at the points of the 3D space at which the determinant of the system of equations becomes zero. An example of such a situation is shown in Figure 5.

An effective way to eliminate such catastrophic errors is to use a cycle of five configurations of the spatial arrangement of BS, that is, to replace the reference BS. If a configuration with some reference BS has the effect shown in Figure 5, then when changing the reference BS, this effect will no longer be present. This method is implemented in clause 6 of the proposed algorithm. All graphs below are calculated using a cycle for five configurations of the spatial arrangement of the BS (Figure 6 and Figure 7).

The RMS area distributions when Gaussian noise is added to the TDoA values with a standard deviation of more than 10 ps have a similar form to Figure 7, and the differences lie in the values of RMS_min_ and RMS_max_, which are summarized in Table 2. RMS_min_ for the linearization method is approximately three times higher than for the analytical method; however, there are gross errors associated with zeroing the determinant of the system at some points in space. Therefore, RMS_min_ and RMS_max_ for the linearization method are not included in Table 1.

As can be seen from the graphs, the accuracy of determining the coordinates strongly depends on the choice of the BS position.

Configurations of global satellite positioning systems are of practical interest. Let us apply the proposed algorithm to simulate the accuracy of determining coordinates in the global positioning system (GPS). Orbiting satellites of the system should be perceived as BS, and the object, the coordinates of which are determined, is on the Earth. We will accept the coordinates of the BS (satellites) as follows: BS*_C_* [*x_C_*; *y_C_*; *z_C_*] = [0; 1.53 × 10^7^; 1.98 × 10^7^], BS*_L_* [*x_L_*; *y_L_*; *z_L_*] = [1.45 × 10^7^; 4.71 × 10^6^; 2 × 10^7^], BS*_R_* [*x_R_*; *y_R_*; *z_R_*] = [8.97 × 10^6^; −1.23 × 10^7^; 2.02 × 10^7^], BS*_U_* [*x_U_*; *y_U_*; *z_U_*] = [−8.97 × 10^6^; −1.23 × 10^7^; 2.04 × 10^7^] and BS*_D_* [*x_D_*; *y_D_*; *z_D_*] = [−1.45 × 10^7^; 4.71 × 10^6^; 2.06 × 10^7^] m, which is presented in Figure 8. The target height is z = 0 m. The number of computational experiments n = 100. The satellites are equipped with atomic clocks with a daily instability of no worse than 10^−13^; therefore, in the calculations, we limited ourselves to the maximum value of the standard deviation of the Gaussian noise TDoA 10 ps (Figure 9 and Figure 10). Such a model can be viewed as a simplified model of the location of satellites and targets on Earth in a GPS positioning system.

RMS_min_ and RMS_max_ for the analytical method and linearization method for the last problem (GPS) are summarized in Table 3.

It can be seen that the proposed algorithm is significantly superior to the widely used positioning algorithm based on the linearization of hyperbolic equations.

## 4. Discussion

The new method, based on the analytical solution of hyperbolic equations, in contrast to many existing methods, makes it possible to obtain an accurate solution for determining the coordinates of an object based on the TDoA measurement. Considerable efforts of researchers have been aimed at finding ways to transform expressions with radicals into linear relations and to ensure fast convergence of solutions. In part, this direction of research is explained by the limited capabilities of the processor technology and the complexity of the implementation of computations of expressions with radicals, which is necessary in the analytical exact method that we have proposed. Of course, the limited mathematical capabilities of inexpensive processors is the main limitation to the widespread implementation of the proposed method. However, the computational element base, both based on processors and FPGAs, is rapidly developing, and we should expect in the near future the appearance of inexpensive processors that allow calculating mathematical functions with radicals in real-time. In addition, in some positioning systems, the function of mathematical processing can be transferred to a system controller (server), in the software of which the proposed method is implemented. In these cases, the restriction on the possibilities of mathematical data processing is removed.

To accurately determine the moment of arrival of a radio signal from an object to base stations, various forms of radio signals are used. The wider the spectrum of the radio signal, the more accurately it is localized in time. Therefore, ultra-wideband (UWB) signals with a wide spectrum have advantages over signals with a shorter spectrum and are used in positioning systems [75]. In our article, the question of the form of the used radio signal is not a subject of research. Radio signal propagation conditions impose a significant influence on the functioning of positioning systems. Multipath propagation, broadband interference, and atmospheric phenomena in global positioning systems lead to TDoA fluctuations. In our work, we did not consider the issues of propagation and reception of radio signals. In our model, the standard deviation of the TDoA is used as a measure of fluctuations, the value of which is determined by the factors listed above.

Within the framework of the adopted model, we have established that the proposed method has the ability to exclude spatial zones of ambiguity in determining the coordinates, which are characteristic of approximate analytical algorithms with one system of equations. The algorithm also eliminates the gross error at the points of the 3D space, at which the determinant of the system of equations becomes zero.

In future work, we propose to calculate the Cramer-Rao Lower Bound (CRLB) for the proposed method, carry out practice implementations to compare the performance with the simulations results, investigate the resistance of the method to non-line-of-sight (NLOS) radio signal propagation, and also consider the applicability of the method for determining the coordinates of a set of objects.

## 5. Conclusions

Modern systems of local and global positioning put forward high requirements for the accuracy of determining the coordinates of objects. This article proposes an analytical method for the exact solution of a system of hyperbolic equations, which, with a high accuracy of less than a few cm, allows one to determine the coordinates of objects and at the same time exclude spatial zones in which approximate analytical algorithms with one system of equations are unable to unambiguously determine the coordinates of an object or give a gross error at the points of the 3D space, at which point the determinant of the system of equations becomes zero. The absence of any omitted terms in the solution provided the developed algorithm with significant resistance to fluctuations in TDoA caused by the conditions of propagation and reception of radio signals and time desynchronization of the BS while maintaining a high accuracy in determining the coordinates. It was found that the proposed method works with high accuracy (5 cm) with BS standard deviation of TDoA in time up to 200 ps, while the linearization method with such BS standard deviation of TDoA provides an accuracy of 60 cm for LPS. The performance of the algorithm has been confirmed by computational experiments for both LPS and GPS.

## Figures and Tables

**Figure 1 sensors-20-05472-f001:**
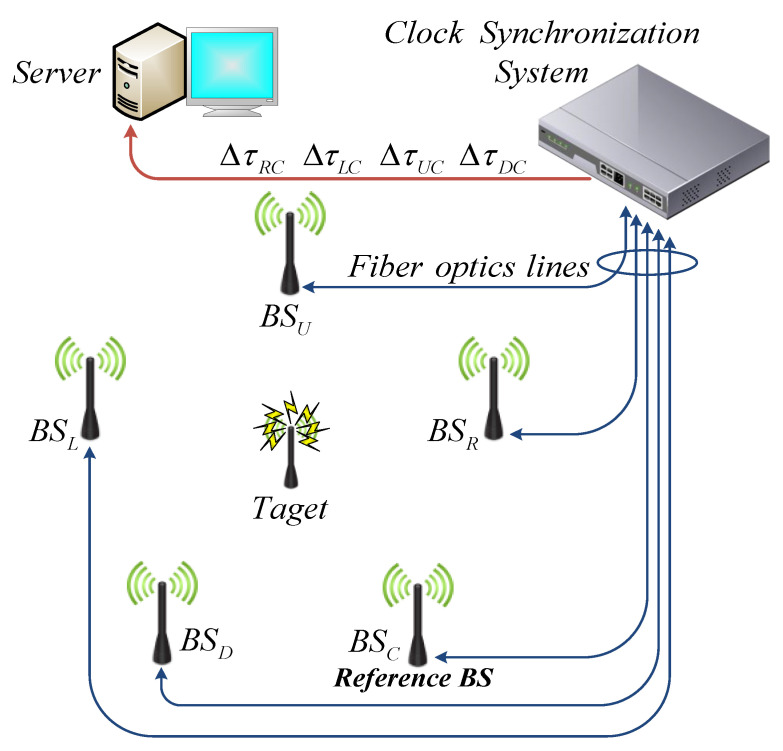
Block diagram of the time difference of arrival (TDoA)-based positioning system with 5 base stations (BSs).

**Figure 2 sensors-20-05472-f002:**
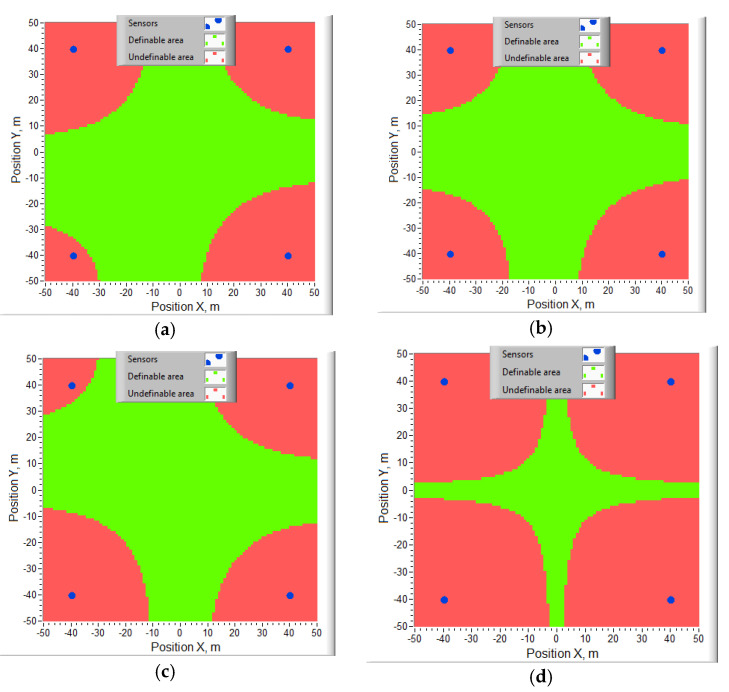
Areas in which the target coordinates are not determined by the algorithm [67]: (**a**) z = 0 m; (**b**) z = 15 m; (**c**) z = 30 m; (**d**) z = 0 m when z*_C,L,R,U_* = 8; 10; 12; 14 m.

**Figure 3 sensors-20-05472-f003:**
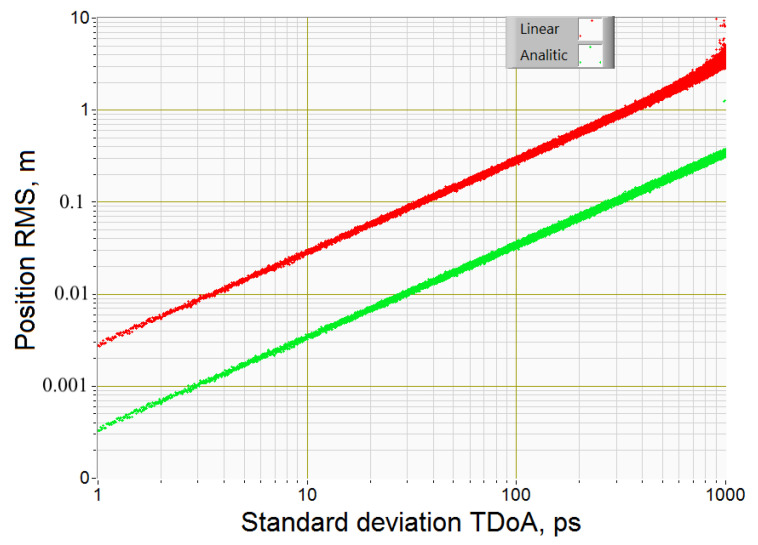
Dependence of the mean square deviation of the object coordinates estimate on the standard deviation of the Gaussian white noise of the difference in arrival time (1 picosecond (ps) = 10^−12^ s): red points for the matrix linearization algorithm, green points for the analytical algorithm proposed.

**Figure 4 sensors-20-05472-f004:**
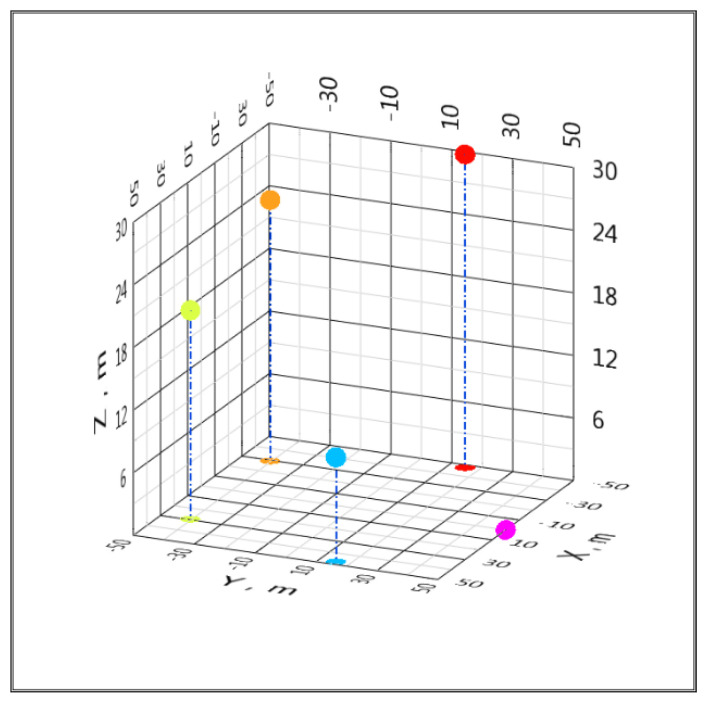
The location of the BS in the example of a local positioning system (LPS) system.

**Figure 5 sensors-20-05472-f005:**
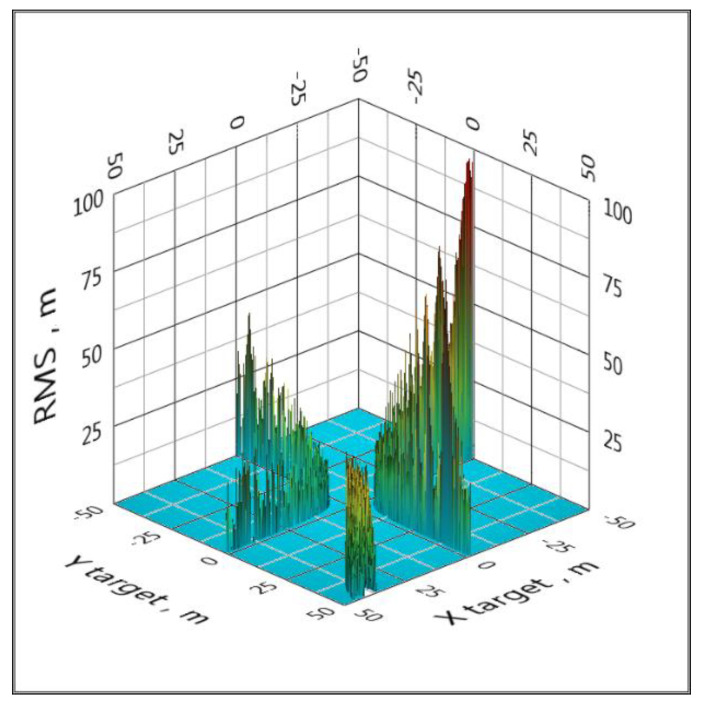
An example of gross errors at the points at which the determinant of the system of equations vanishes (LPS).

**Figure 6 sensors-20-05472-f006:**
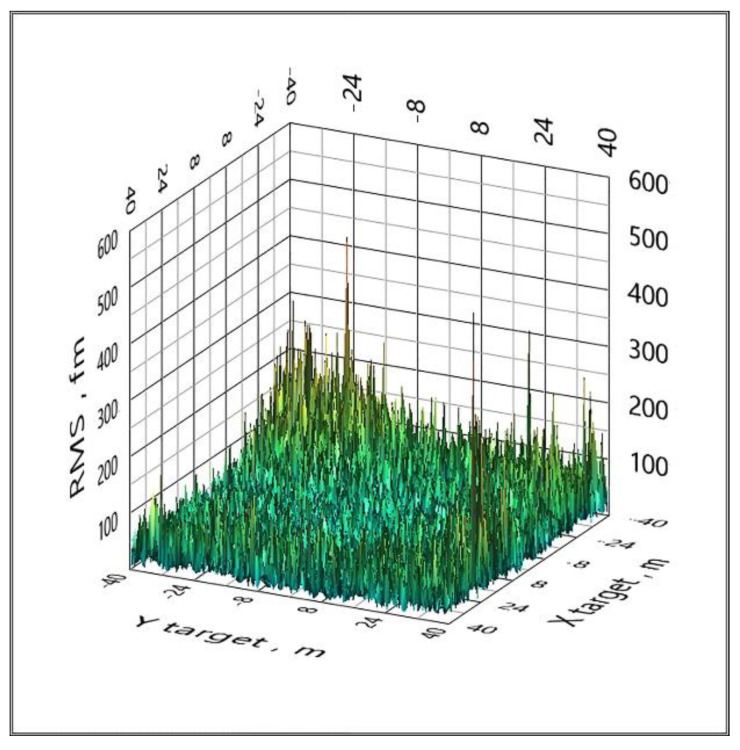
The distribution of the RMS (1 femtometer (fm) = 10^−15^ m) coordinate estimate over the area when Gaussian noise is added to the TDoA values with a standard deviation of 0 ps (LPS).

**Figure 7 sensors-20-05472-f007:**
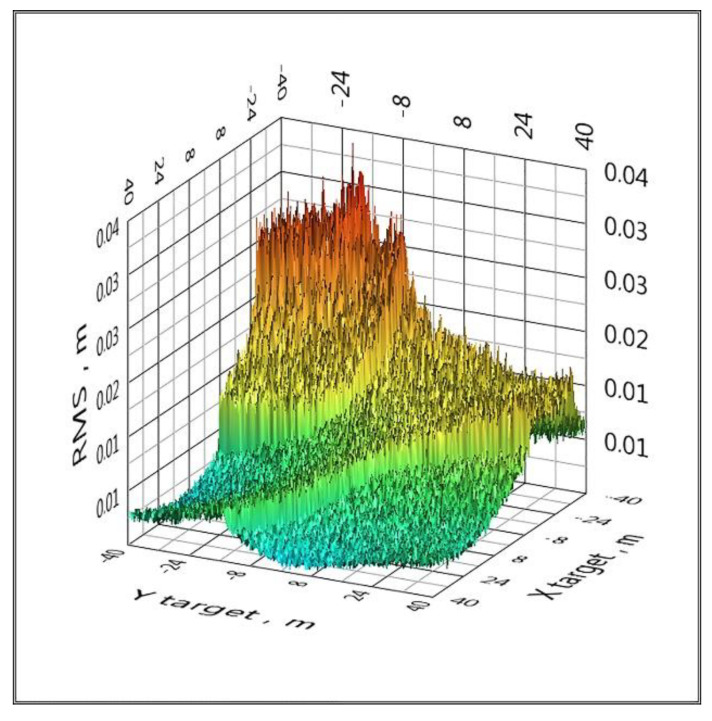
The distribution of the RMS coordinate estimate over the area when Gaussian noise is added to the TDoA values with a standard deviation of 10 ps (LPS).

**Figure 8 sensors-20-05472-f008:**
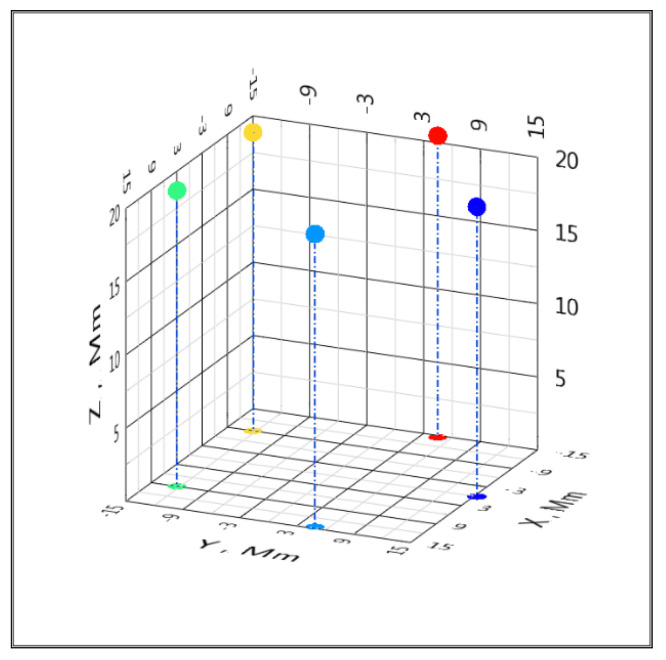
Simplified model of the location (1 megameter (Mm) = 10^6^ m) of satellites (red) and a target on Earth (blue) in a GPS positioning system.

**Figure 9 sensors-20-05472-f009:**
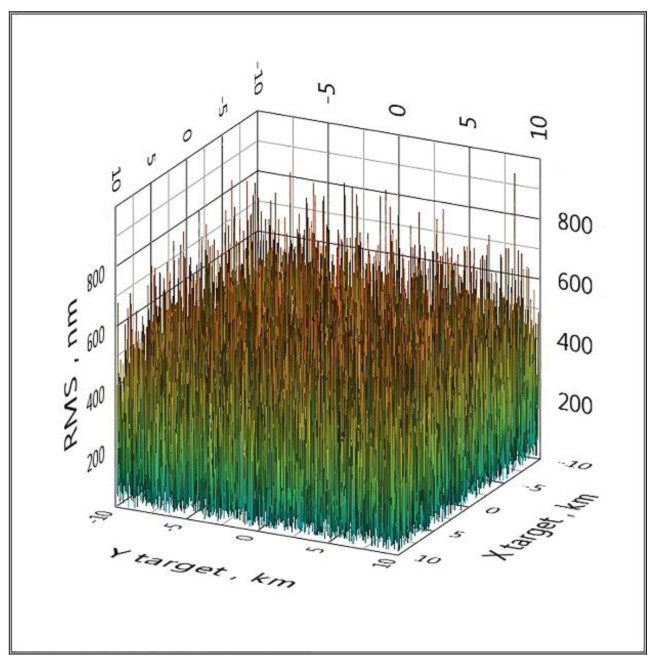
The distribution of the RMS coordinate (1 nanometer (nm) = 10^−9^ m) estimate over the area when Gaussian noise is added to the TDoA values with a standard deviation of 0 ps (GPS).

**Figure 10 sensors-20-05472-f010:**
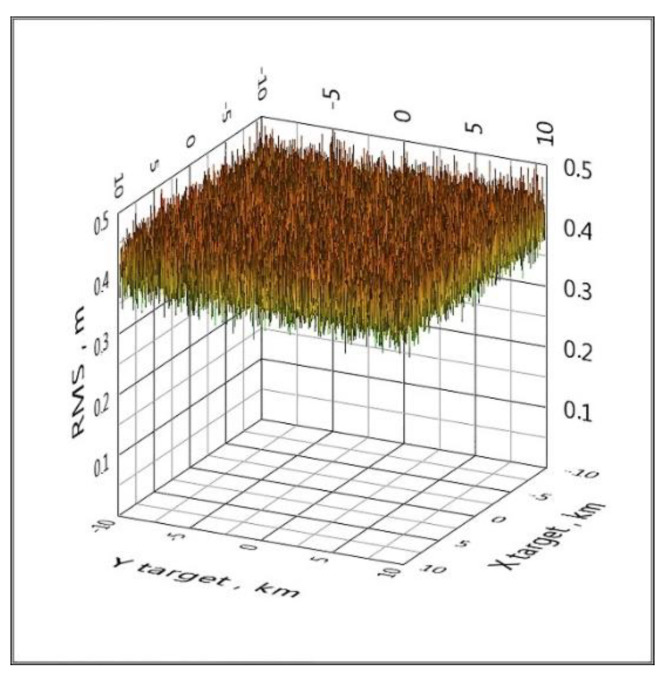
The distribution of the RMS coordinate estimate over the area by adding Gaussian noise to the TDoA values with a standard deviation of 10 ps (GPS).

**Table 1 sensors-20-05472-t001:** Replacement of the notation.

*i* = 1		*i* = 2		*i* = 3		*i* = 4	
L	→	R	→	U	→	D	
R	→	U	→	D	→	L	
U	→	D	→	L	→	R	
D	→	L	→	R	→	U	
E	→	F	→	G	→	H	
F	→	G	→	H	→	E	
G	→	H	→	E	→	F	

**Table 2 sensors-20-05472-t002:** RMS_min_ and RMS_max_ for analytical method (LPS).

Deviation TDoA, ps	RMSmin, m	RMSmax, m
0	0	5.56 × 10^−13^
10	0.0032	0.037
100	0.031	0.338
1000	0.328	3.3

**Table 3 sensors-20-05472-t003:** RMS_min_ and RMS_max_ for analytical and linear methods (GPS).

Deviation TDoA, ps	RMSmin, m. Analytical Method	RMSmin, m. Linear Method	RMSmax, m. Analytical Method	RMSmax, m. Linear Method
0	0	1.91 × 10^−9^	1.013 × 10^−6^	8.80 × 10^−5^
1	0.029	0.166	0.055	0.318
10	0.29	1.67	0.552	3.17

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
