# Peer review of "Method for Remote Determination of Object Coordinates in Space Based on Exact Analytical Solution of Hyperbolic Equations"

_sensors, 2020, doi:10.3390/s20195472_

Round 1

Reviewer 1 Report

Authors talk about the development of hyperbolic solution algorithm, which is "tens of times higher than the method of linearization of hyperbolic equations and is less sensitive to time desynchronization". Declared contribution is highly doubtful. Comments:

  1. One of the main measures to assess any proposed positioning solution algorithm is CRLB or ML solution. There is no such comparison in this work, thus it is hard to conclude about any contribution. Moreover, comparison with linearization of hyperbolic equations approach, proposed by W. H. Foy in his 1976 paper “Position-location solutions by Taylor-series estimation,” seems to be not up to date.
  2. For the last three decades the was a vast majority of papers devoted to TDOA solution. World-known results were developed by team of Y.T.Chan, K.S.Ho, Wenwei Xu, Xiaoning Lu, Le Yang, . Sun et. al. Among them there are well known papers: "A simple and efficient estimator for Hyperbolic location" (1994), "On the Use of a Calibration Emitter for Source Localization in the Presence of Sensor Position Uncertainty," (2008), "Bias Reduction for an Explicit Solution of Source Localization Using TDOA" (2012), "Solution and Analysis of TDOA Localization of a Near or Distant Source in Closed Form," (2019). Authors should compare their solution with some well known-results. Equations (1)-(13) of the paper under consideration seems to be trivial and unmatured comparing to above-listed example works.
  3.  Authors investigate 3D location estimation, however, present 2D "significant areas" in Fig. 2 with fixed z. Why not well known metric GDOP, based on CRLB, is used instead?
  4. Figures 5-10 for RMS in 3D looks poorly. It might be better to give 2D contout plots.
  5.  In the conclusion authors state, that "compared to previous publications, the proposed solution is more computationally attractive". How it can be verified, where are any quantitative measures? Moreover it is stated, that "proposed algorithm is promising for ... unmanned vehicles in smart city applications", however there is neither requirements, nor any specific parameters for such applications throughout the paper. 
  6. Introduction and reference list has enough volume, but it's content leaves no opportunity to perceive any reasonable contribution in the topic under consideration. 
  7. Stylistics seems weak. TDoA abbreviation, for example, disclosures several times.

Author Response

Thank you for your comments

Our answers for your comments are following:

  1. One of the main measures to assess any proposed positioning solution algorithm is CRLB or ML solution. There is no such comparison in this work, thus it is hard to conclude about any contribution. Moreover, comparison with linearization of hyperbolic equations approach, proposed by W. H. Foy in his 1976 paper “Position-location solutions by Taylor-series estimation,” seems to be not up to date.

Answer: The Cramer-Rao Lower Bound (CRLB) or maximum likelihood estimator (MLE) is one of the methods comparison criteria. The CRLB identifies the best potential of the method, but does not answer the question of how this potential can be achieved. Therefore, we have chosen another criterion that characterizes the real operation of the algorithm - the root-mean-square deviation of the object coordinates.

W. H. Foy in his 1976 paper “Position-location solutions by Taylor-series estimation” developed an iterative method for solving hyperbolic equations, which is characterized by slow convergence and a significant influence of the choice of the initial point of iterative procedures. At present, W. H. Foy's iterative method is not used. By Chan and Ho proposed a non-iterative solution to the problem of estimating the hyperbolic position, used two weighted least-squares (WLS) to solve the linear equations and gave an explicit solution with high location accuracy. Currently, the Chan method is significantly developed, and we use in the paper a comparison with the improved method of Chan.

  1. For the last three decades the was a vast majority of papers devoted to TDOA solution. World-known results were developed by team of Y.T.Chan, K.S.Ho, Wenwei Xu, Xiaoning Lu, Le Yang, . Sun et. al. Among them there are well known papers: "A simple and efficient estimator for Hyperbolic location" (1994), "On the Use of a Calibration Emitter for Source Localization in the Presence of Sensor Position Uncertainty," (2008), "Bias Reduction for an Explicit Solution of Source Localization Using TDOA" (2012), "Solution and Analysis of TDOA Localization of a Near or Distant Source in Closed Form," (2019). Authors should compare their solution with some well known-results. Equations (1)-(13) of the paper under consideration seems to be trivial and unmatured comparing to above-listed example works.

Answer: The reviewer cited 4 publications of Ho K.C. as an example. 1994, 2008, 2012 and 2019. Initially, in our article in the list of references there are 5 articles by Ho K.C. 1994, 2012, 2014, 2017 and 2020, of which two articles coincide with those given by the reviewer. We have additionally included in our article in the literature review an article recommended by the reviewer: "Solution and Analysis of TDOA Localization of a Near or Distant Source in Closed Form," (2019).

Comparison with results from other authors is added to new section 4 – Discussion

Equations (1) - (13) really represent the solution of elementary equations and can be used directly by the hardware developers. We have given the final expressions in full so that they are directly applicable not only by mathematicians, but also to programmers. This will expand the scope of readers of the paper.

3.Authors investigate 3D location estimation, however, present 2D "significant areas" in Fig. 2 with fixed z. Why not well known metric GDOP, based on CRLB, is used instead?

The term GDOP means "geometric dilution of precision". As the name suggests, the term GDOP is the most general one. Of all the articles recommended by the reviewer, this term appears only once in one article. Other articles rarely use the term GDOP. Therefore, the term "metric GDOP" should not be considered widely known. We use the standard deviation of the object position (RMS) - a term that is really widely known and used throughout.

  1. Figures 5-10 for RMS in 3D looks poorly. It might be better to give 2D contout plots.

Figures 5,6,7,9,10 are statistical in nature, as a result of which their 2D contour plots are blurred, making it difficult to perceive the RMS values. Better, after all, to leave the drawings in 3D format.

  1. In the conclusion authors state, that "compared to previous publications, the proposed solution is more computationally attractive". How it can be verified, where are any quantitative measures? Moreover it is stated, that "proposed algorithm is promising for ... unmanned vehicles in smart city applications", however there is neither requirements, nor any specific parameters for such applications throughout the paper. 

This phrase "compared to previous publications, the proposed solution is more computationally attractive" has been removed from the conclusion. We agree with the reviewer.

Explanations for the statement "proposed algorithm is promising for ... unmanned vehicles in smart city applications" are presented in new Section 4 Discussion

  1. Introduction and reference list has enough volume, but it's content leaves no opportunity to perceive any reasonable contribution in the topic under consideration

The introduction the current state of the application of methods and algorithms for calculating coordinates in positioning systems have been analyzed . Anyway most systems use approximate methods for estimating the coordinates of an object, based on discarding high-order terms of expansion in a series in a small parameter. Only a few papers offer methods based on the rigorous solution of hyperbolic equations. These methods require development. In our paper, an analytical method for determining the position of a target based on the analysis of Time Difference of Arrival (TDoA) of microwave radar signals from the transmitter to Base Stations (BS) receivers is proposed. The main feature of the method is that in can eliminate the ambiguity in determining of 3D coordinates of a target and improves the accuracy of determining coordinates when the TDoA measurements are not synchronized with BS.

  1. Stylistics seems weak. TDoA abbreviation, for example, disclosures several times

Eliminated the disclosure of the abbreviation TDoA in several places of the text, the style of the paper have been improved.

Reviewer 2 Report

Indoor positioning is becoming very hot in recent years. But due to cost consideration, IR-UWB is very polupar for TDoA system. Please add some comments comparing your system to IR-UWB TDoA system.

Author Response

Thank you for your comment

Our answers for your comment is following:

Indoor positioning is becoming very hot in recent years. But due to cost consideration, IR-UWB is very polupar for TDoA system. Please add some comments comparing your system to IR-UWB TDoA system.

To accurately determine the moment of arrival of a radio signal from an object to base stations, various forms of radio signals are used. The wider the spectrum of a radio signal, the more accurately it is localized in time. Therefore, the so-called UWB (ultra wide band) signals, which have a wide spectrum, have advantages over signals with a shorter spectrum. In our article, the question of the form of the used radio signal is not a subject of research. It is believed that the time of arrival of a radio signal to base stations fluctuates due to the propagation conditions in space and the desynchronization of the base station equipment. In the paper we use as a measure of fluctuations a standard deviation of the TDoA.

Reviewer 3 Report

1) The literature is very good. However, I suggest you mention recent works especially for indoor and AoA localization, such as

-https://doi.org/10.1109/ICCE46568.2020.9043072

-https://doi.org/10.1109/TCSII.2020.2995064

- https://doi.org/10.1109/TCSI.2020.2979347

-https://doi.org/10.3390/s17122917

-https://doi.org/10.1109/APMC46564.2019.9038530

2) Is the equation (2) derived from the equation (1). It is not clear the starting point

3) fix the position of the figure 4 caption

4) fix the position of the figure 10 caption

5) My big concern is about most of the hypothesis made by the Authors that are unrealistic and, as such, unacceptable. Many well-known RF issues, as multipath, wide-band interferences and so on, are completely ignored, but these issues heavily affect every RF localization problem!

6) Did the authors implement the proposed system in a real environment? Please add at least the limitation of this work and the possible effect of the multipath on such a solution.

Author Response

Thank you for your comments

Our answers for your comments are following:

  1. The literature is very good. However, I suggest you mention recent works especially for indoor and AoA localization, such as

-https://doi.org/10.1109/ICCE46568.2020.9043072

-https://doi.org/10.1109/TCSII.2020.2995064

- https://doi.org/10.1109/TCSI.2020.2979347

-https://doi.org/10.3390/s17122917

-https://doi.org/10.1109/APMC46564.2019.9038530

Thanks to the reviewer for links to interesting articles on our topic. We added them to Introduction and references

  1. Is the equation (2) derived from the equation (1). It is not clear the starting point

Expression (2) is a consequence of expression (1) after carrying out these mathematical operations. To eliminate the ambiguity indicated by the reviewer, a link to formula (1) has been added to the revised version of the article.

  1. fix the position of the figure 4 caption

The figure caption has moved to the next page according automatically formatting. We have corrected this.

   4 fix the position of the figure 10 caption

The figure caption has moved to the next page according automatically formatting. We have corrected this.

  1. My big concern is about most of the hypothesis made by the Authors that are unrealistic and, as such, unacceptable. Many well-known RF issues, as multipath, wide-band interferences and so on, are completely ignored, but these issues heavily affect every RF localization problem!

In new section 4 Discussion we have added discussions of the RF localization issues.

  1. Did the authors implement the proposed system in a real environment? Please add at least the limitation of this work and the possible effect of the multipath on such a solution.

In new section 4 Discussion we have added discussions of the RF localization issues.

Round 2

Reviewer 1 Report

Manuscript has been improved and reasoning became more convincing. 

Authors answers for comments proves understanding of current weak parts

and satisfies reviewer for correct direction of future work.

Reviewer 3 Report

The authors have addressed all my considerations. Thanks